



**Retrieval of cloud properties from sky radiometer observed**
**spectral zenith radiances**
Pradeep Khatri[1], Hironobu Iwabuchi[1], Tadahiro Hayasaka[1], Hitoshi Irie[2], Tamio Takamura[2],
Akihiro Yamazaki[3], Alessandro Damiani[2], Husi Letu[4], Qin Kai[5]
[1]Center for Atmospheric and Oceanic Studies, Graduate School of Science, Tohoku University, Sendai, Japan
[2]Center for Environmental Remote Sensing, Chiba University, Chiba, Japan
[3]Meteorlogical Research Institute, Tsukuba, Japan
[4]Institute of Remote Sensing and Digital Earth, Chinese Academy of Sciences, Beijing, China
[5]School of Environment and Geoinformatics, China University of Mining and Technology, Xuzhou, China
*Correspondence to*: Pradeep Khatri (pradeep.khatri.a3@tohoku.ac.jp)
**Abstract:** An optimal-estimation based algorithm to retrieve cloud optical thickness (COD) and cloud-particle effective
radius (CER) from spectral zenith radiances observed by a narrow field of view (FOV) ground-based sky radiometer is
developed. To further address the filter-degradation problem while analyzing data of long-term observation, an on-site
calibration procedure is proposed, which is found to have a very good accuracy with respect to a standard procedure,
i.e., a procedure of deriving calibration constants using a master instrument. An error evaluation study conducted by
assuming errors in observation-based transmittances and ancillary data of water vapor concentration and surface albedo
suggests that the errors in input data can influence retrieved CER more effectively than COD. Except for some narrow
domains that fall within COD < 15, the retrieval errors are very small for both COD and CER. The retrieved cloud
properties are found to reproduce the broadband radiances observed by a narrow FOV radiometer more precisely than
broadband irradiances observed by a wide FOV pyranometer, justifying the quality of retrieved product (at least COD)
and, at the same time, indicating the important influence of instrument FOV in cloud remote sensing. Further, CODs
(CERs) between sky radiometer and satellite observations show fairly good (poor) agreement.

**1   Introduction**
Clouds play an important role to drive the climate system and hydrological cycle (Rosenfeld et al., 2014). Their
accurate representation in the global climate model remains one of the largest uncertainties (Foster et al., 2007). Clouds
are being observed from the space by using various sensors onboard various satellites, which are greatly helping the
scientific community to understand more about cloud characteristics and their roles on climate system, hydrological
cycle, and so on. Despite such profound scientific implication, the quality assurance of such satellite observed cloud
properties has always become an important task in the field of cloud remote sensing. It has been recognized as a
challenging task primarily due to the lack of enough standard data representing different atmospheric conditions to





perform quality assessment. Unlike advancement in aerosol observation through surface networks such as AERONET
(https://aeronet.gsfc.nasa.gov/) and SKYNET (http://atmos3.cr.chiba-u.jp/skynet/), clouds are being observed routinely
from the surface at a very limited stations, and most of those observation data are not easily accessible. Taking into
account of multiple stations of above-mentioned ground-based networks around the world, cloud observations from
them should be strengthen as their operating instruments for aerosol observation are capable for cloud observation as
well. This can greatly help satellite remote sensing community to validate the cloud products and the whole cloud
research community to study clouds in more detail by using high resolution surface data. This recognition strongly
motivated this study.

42        It is important to mention that efforts have been made in the past for studying clouds from the surface by using

zenith radiances observed by radiometers belonging to AERONET (e.g., Chiu et al., 2010, 2012) and SKYNET (e.g.,
Kikuchi et al., 2006). In accordance with the literature, the radiometers belonging to AERONET and SKYNET are
termed as sun photometer and sky radiometer, respectively from here. Similar to space-based cloud remote sensing
using reflected signals (e.g., Nakajima and King, 1990), those studies using data of sun photometer and/or sky
radiometer are based on a framework of a look-up-table (LUT) use. The fundamental idea is to tally observed signals
with LUT of signals corresponding to priory known cloud optical depth (COD) and cloud-particle effective radius
(CER). This signal can be zenith radiance or transmittance. Chiu et al. (2010) retrieved COD from LUT of zenith
radiances of water non-absorbing wavelengths constructed by assuming a fixed CER; Chiu et al. (2012) and Kikuchi et
al. (2006) used LUT of transmittances of water non-absorbing and absorbing wavelengths to infer COD and CER
simultaneously. It is important to note that the reflected signals for water non-absorbing and absorbing wavelengths can
have nearly one-to-one relationships with COD and CER, respectively. On the other hand, such behaviors cannot be
seen for transmitted signals, making the retrieval process very difficult for LUT based approach using transmitted
signals. In addition, unlike reflected signal, the transmitted signal is weakly sensitive to CER change. This again adds
complexity in the retrieval while using transmitted signals. Further, the shape of LUT is subject to change depending on
the change of solar position, making the retrieval process more cumbersome, if LUTs developed for a limited number of
specific solar positions are used. To overcome such difficulties, some innovative techniques have been proposed in the
past. For example, MCBride et al. (2011) developed a spectral method by using the slope of the transmittances of 13
wavelengths in between 1565nm and 1634nm and transmittance at the visible wavelength of 515nm to retrieve COD
and CER simultaneously. LeBlanc et al. (2015) derived 15 parameters to quantify spectral variations in shortwave
transmittances due to absorption and scattering of liquid water and ice clouds, manifested by shifts in spectral slopes,
curvatures, maxima, and minima to discriminate cloud phase and retrieving COD and CER. Unfortunately, those
techniques developed for radiometers of high spectral resolution are less suitable for both sun photometer and sky
radiometer because they have a very limited number of channels.

66        Taking above-mentioned difficulties in mind, we develop a retrieval algorithm based on optimal-estimation

method, i.e., a maximum a posteriori (MAP) method (Rodgers, 2000). We use three carefully selected wavelengths to
retrieve COD and CER simultaneously. Further, an on-site calibration method is proposed to cope the filter degradation



problem while analyzing data of long-term observation. Though the algorithm is developed by using data of sky
radiometer, it is equally applicable for data of sun photometer.

71       This study is designed in the following way: A brief description of sky radiometer is given in section 2. The

methodology, retrieval error, and quality assessment of retrieved products are discussed in sections 3, 4 and 5,
respectively. Finally, the conclusion is presented in section 6.

**2    Sky radiometer**
The sky radiometer (Model: POM-02), manufactured by PREDE Co. Ltd., Japan, is an instrument capable to make
observations of direct intensity, angular sky radiance (both almucantar and principle plane scans), and zenith sky
radiance at 11 wavelengths at specified time interval. The field of view (FOV) is 1°. The most commonly used
wavelengths by SKYNET are 0.315, 0.34, 0.38, 0.4, 0.5, 0.675, 0.87, 0.94, 1.02, 1.627, and 2.2 μm. Among them, the
direct and angular sky radiances at the wavelengths of 0.34, 0.38, 0.4, 0.5, 0.675, 0.87, 0.94, and 1.02 μm, at which the
absorptions by atmospheric gases and water/ice are negligible, are in use for aerosol remote sensing (Nakajima et al.,
1996; Hashimoto et al., 2012); the direct intensities observed at the wavelengths of 0.315 μm and 0.94 μm are in use for
remote sensing of ozone (Khatri et al., 2014) and water vapor (e.g., Campanelli et al., 2014), respectively. The zenith
sky radiances have different potential applications from different perspectives. Currently, the zenith sky radiances of
cloudy skies have been used for cloud remote sensing (e.g., Kikuchi et al., 2006). The calibration constant terms for sky
radiance (both angular and zenith) and direct intensity are required while deriving physical data from observation
signals via retrieval algorithms. One of the largest benefits of PREDE sky radiometer is that those calibration constants
can be obtained from field observation data themselves as outlined by Nakajima et al. (1996). In brief, an Improved
Langley (IL) method (Nakajima et al., 1996; Campanelli et al., 2004), which is an alternation of Normal Langley (NL)
method, can be used to obtain calibration constants for direct intensities. Similarly, the solar disk scan method, which
can be alternation of using integrating sphere etc., can be used to determine the calibration constant for sky radiances. A
more detail about sky radiometer and its calibration can be found in Khatri et al. (2016).

**3    Methodology**
The schematic diagram of the study method is shown in Fig. 1. We use sky radiances (E) observed at three longer
wavelengths (0.87, 1.02, and 1.627 μm) excluding 2.2 μm. The longest wavelength of 2.2 μm is not used considering
the application of proposed algorithm for sun photometer data as well because the longest wavelength used by
AERONET is 1.64 μm. Similarly, the shorter wavelengths are not used to avoid the effect of aerosols as far as possible.
The observed E can be converted to the transmittance (T) by the equation below:
$$T(\lambda) = \frac{\pi E(\lambda)}{\mu_0 \Delta\Omega(\lambda) F_0(\lambda)},$$    (1)
where $\mu_0$ is the cosine of the solar zenith angle, $\Delta\Omega$ is the calibration constant for sky radiance, which is also termed as
solid view angle in SKYNET community, $F_0$ is the calibration constant for direct intensity, and $\lambda$ is the wavelength. $\Delta\Omega$



for those three wavelengths can be determined from the solar disk scan of very clear sky days (Nakajima et al., 1996).
Though the current IL method can be used to determine temporal $F_0$ for the first two wavelengths (0.87 and 1.02 μm), it
is less applicable for water absorbing wavelength, such as 1.627 μm. For 1.627 μm, one may use $F_0$ derived from NL
method, but NL is less practical to be conducted routinely within short time interval (e.g., each month) to derive
temporal $F_0$. We prefer to use temporal $F_0$ for all wavelengths to take into account of filter degradation with time (e.g.,
Khatri et al., 2014). To derive temporal $F_0$ at 1.627 μm, we use an alternative method of IL as proposed by Khatri et al.
(2014). In brief, aerosol data (refractive index and volume size distribution) and direct intensity observed at 1.627 μm
($F_{1.627}$) are used. Note that aerosol optical thickness ($\tau_{aer}$) depends primarily on aerosol size distribution, and the
refractive index has a very mere contribution to $\tau_{aer}$ (King, 1978; Khatri and Ishizaka, 2007). Thus, the refractive index
at the wavelength of 1.02 μm, which is the highest wavelength for routine aerosol retrieval, is assumed to be same for
1.627 μm while calculating $\tau_{aer}$ at 1.627 μm from volume size distribution using Mie calculation. The optical air mass
($m$) and sun-earth distance ($R$) are calculated from latitude and longitude of observation site and observation time.
Similarly, the Rayleigh-scattering optical depth at 1.627 μm ($\tau_{Ray,1.627}$), though very small in magnitude, is calculated
from the atmospheric pressure of observation site. Finally, the beer-lambert law of $ln(F_{1.627}R^2) = lnF_{0,1.627} -$
$(\tau_{aer} + \tau_{Rayleigh})m$ is used to determine $lnF_{0,1.627}$, i.e., the natural logarithm of the calibration constant of direct intensity at
1.627 μm. It is done by using data of all clear sky periods of each month by correlating $ln(F_{1.627}R^2)$ with $(\tau_{aer} + \tau_{Rayleigh})m$.
The outlier that deteriorates the correlation utmost is detected and removed in each iteration till the condition of
correlation coefficient ($r^2$) ≥ 0.995 is satisfied. To understand the quality of thus calculated $lnF_{0,1.627}$ values, we
compare them with data determined from independent standard method. The standard method refers to the method of
deriving calibration constant by performing collocated observation of a field and master instruments together. In Fig. 2,
the comparison is shown for three different sky radiometers operated at the observation sites of Hedo-misaki (26.87°N,
128.25°E), Fukue-jima (32.75°N, 128.68°E), and Sendai(38.26°N,140.84°E). Figure 2 shows a very good agreement
between our method and the standard method for all three sky radiometers. Figure 2 also shows a relative difference.
The relative difference (in percentage) is defined as the difference between our method and the standard method
normalized by the value of the standard method. The relative difference can be seen to be less than 0.05% for all sky
radiometers. This confirms the soundness of proposed method, which is not only inexpensive, but also very easy. Thus,
the proposed method can be used to determine temporal $lnF_{0,1.627}$, which is very useful while analyzing data of
long-term observation by coping filter degradation problem. By using volume size distribution and refractive indices of
respective wavelengths, the proposed method can be used for first two wavelengths as well. There is negligible
difference in the values obtained between IL method and this method for the first two wavelengths. This study uses the
values obtained from the proposed method for all wavelengths to avoid the difficulty of reading $lnF_0$ from different
files.

135          Along with $T$ values of three wavelengths obtained from Eq. (1), we use precipitable water content (PWC) and

spectral      surface      albedo      data.      They      are      obtained      from      radiosonde      observation



(http://weather.uwyo.edu/upperair/sounding.html) and MODIS observation (product name: MCD43A4), respectively.
Finally, COD and CER are retrieved simultaneously by minimizing the cost function ($J$) below:
$J = (x - x_a)^T S_a^{-1}(x - x_a) + [y - F(x, b)]^T S_y^{-1}(y - F(x, b))$ ,      (2)
where $x$ is a state vector, $x_a$ is an a priori vector, $S_a$ and $S_y$ are error covariance matrices for the a priori and
measurement, respectively, $y$ is the measurement vector, $F$ is the forward model, and $b$ is the model parameter vector
(ancillary data) . The terms $x, y$, and $b$ are defined as:
$x = \begin{pmatrix} ln\tau \\ lnr_e \end{pmatrix}$, $y = \begin{pmatrix} lnT_{1.627} \\ lnT_{1.02} \\ lnT_{0.87} \end{pmatrix}$, and $b = \begin{pmatrix} W \\ A_{1.627} \\ A_{1.02} \\ A_{0.87} \end{pmatrix}$,
where $\tau$ and $r_e$ represent COD and CER, respectively; $W$ and $A_\lambda$ represent PWC and surface albedo at wavelength $\lambda$,
respectively. Both $S_a$ and $S_y$ are assumed to be diagonal matrices. $x_a$ and the diagonal elements of $S_a$ are determined
from one-year data of water cloud properties observed over Japanese SKYNET sites by Advanced Himawari Imager
(AHI) sensor onboard Himawari-8, a Japanese geostationary satellite. The diagonal terms for $S_y$ are determined based
on simulation of perturbations in $T(\lambda)$ generated from 300 random gaussian noises of error sources discussed in section
4. The Santa Barbara DISORT Atmospheric Radiative Transfer (SBDART) model (Ricchiazzi et al., 1998) is used for
forward modelling, and the Levenberg-Marquardt method is used to minimize the cost-function.

## 152     4    Retrieval error

In order to understand the performance of the proposed algorithm for different types of input data (transmittance and
ancillary data), retrieval errors are calculated by assuming errors on them. The retrieval errors are calculated for COD of
1 - 64 and CER of 2 − 32 μm at an interval of 1 for each. The simulations are performed for solar zenith and azimuth
angles of 30° and 0°, respectively by assuming cloud phase as water cloud. We assume 1% error in $lnF_0(\lambda)$, which is
significantly larger than the maximum error in $lnF_0(\lambda)$ shown in Fig. 2 and discussed in section 3. This large error in
$lnF_0(\lambda)$ is assumed to incorporate errors in $T(\lambda)$ generated from also other possible factors, such as radiance
measurement and $\Delta\Omega(\lambda)$ estimation. Similarly, we assume surface albedo of 0.15 for all three wavelengths and PWC of
1.0 cm by assuming errors on them as ±0.025 and 1.0 cm, respectively. Though $F_0(\lambda)$ in actual data analysis is the
instrument signal equivalent to the measurement performed at the top of the atmosphere (TOA), it is the incident
irradiance at TOA (unit: W/m$^2$/nm) calculated from the radiative transfer model for error evaluation simulations
discussed in this section. For each set of priory known COD and CER, 100 random gaussian noises for each error
source mentioned above are added in the retrieval to simulate 300 sets of COD and CER. Out of them, the successful
retrievals ($J \leq 3$) are used to calculate the mean bias error (MBE) as below:
$MBE = \frac{\sum_{i=1}^{n}(\frac{Si}{Tr} - 1)}{n}$      (3),
where, $Si$ and $Tr$ represent the simulated and true values, respectively. Only the MBE is discussed here because the
error map evaluated in other form, such as root mean square error (RMSE), contains the same qualitative information.



Figure 3 presents MBE for COD (first column), MBE for CER (second column), and total number of successful
retrieval (third column) for each type of error separately and in combination. Figures 3(a) – 3(c), 3(d) – 3(f), 3(g) – 3(i),
and 3(j) – 3(l) correspond to errors in transmittance, surface albedo, PWC and all of them, respectively. The 100%
unsuccessful retrieval is shown by black color. One can note that the retrieval can be more uncertain mainly when COD
is less than ~15. Further, regardless of the error source, the retrieval error is relatively high especially for small (CER <
~7 µm) and large ( CER > ~13 µm) cloud droplets. In general, the error domains of CER are expanded by overlapping
the error domains of COD. This suggests that the error in input data can affect CER retrieval more effectively than
COD retrieval. Among three different error sources, the error in transmittance can dominate the effect of remaining two
error sources. The successful retrieval number corresponding to each error source shown in the third column clearly
suggests two domains where the algorithm finds difficulty to fit the measurement based transmittances with modeled
values. Those domains exist in ~8 < COD < ~16 with CER > ~13 µm and CER < ~7 µm. Those domains have relatively
high retrieval errors as shown in the first and second columns. The relatively high errors in COD and CER are further
extended for COD < ~8 despite enough number of successful retrievals. The contour lines for $T(\lambda)$ shown in Figs. 4(a),
4(b), and 4(c) for the wavelengths of 0.87, 1.02 and 1.627 µm, respectively can help to shed light for understanding
those domains. The $T(\lambda)$ values shown in Figs. 4(a) – 4(c) correspond to the condition of no error in input data. First
talking about unsuccessful retrievals noted for ~8 < COD < ~16 and CER > ~13 µm domain, Figs. 4(a) - 4(c) suggest
that $T(\lambda)$ values can hardly change with the increase of CER when CER > ~13 µm. As a result, the CER retrieval greater
than ~13 µm is very uncertain and the retrieved CER is generally underestimated. Note that $T(\lambda)$ contour lines falling
within ~8 < COD < ~16 get appear again for COD < ~2. Therefore, in an attempt of searching the best set of COD and
CER by trying to fit the inputted $T(\lambda)$ values with the modeled values, the algorithm can mistakenly search the plausible
solution from this small COD domain. If this happens, the retrieval may not confine within $J \leq 3$. The algorithm is
likely to compensate such underestimated CERs by overestimating CODs as clearly shown by Figs. 3(a) - 3(b) or 3(j) -
3(k). Similarly, talking about failed retrievals for CER < ~7 µm, a very non-uniform change of $T(1.627\ \mu m)$ associated
with CER change, as shown by Fig. 4(c), can be the important factor. Such very non-uniform response of CER towards
$T(1.627\ \mu m)$ change can mislead the algorithm while searching the best set of COD and CER and/or may force the
algorithm to mistakenly shift to COD < ~2 domain to search the plausible solution. A very careful look suggests that
both CER and COD are overestimated for CER > ~7 µm. Despite enough number of successful retrieval, one can note
relatively high errors in retrieved values for COD < ~8. Similar to above-discussed error domains, the retrieval errors
are mainly confined for relatively large and small values of CER. It is important to note that the peak values of $T(\lambda)$
generally fall for ~3 ≤ COD ≤ ~6. In other words, the competition between forward scattering and absorption is
maximum to increase or decrease $T(\lambda)$ within this COD range. Note that the absorption tends to reduce $T(\lambda)$, whereas
the forward scattering tends to increase it. Not only COD, but also CER is equally important to increase or decrease
$T(\lambda)$, and the algorithm needs to take into account of both COD and CER changes while searching the plausible set of
COD and CER. Thus, it is the most sensitive COD range for the ambiguous solution of COD and CER in transmittance
based remote sensing. Therefore, even a small degree of error in input data can divert both COD and CER significantly



from their true values. Though weak, this phenomenon can be still active in the vicinity of this COD range to bring
error in retrieved values even for COD < ~3. The weak CER response towards $T(\lambda)$ for large CERs, as discussed above,
again plays an important role to bring errors in retrieved values for relatively large CERs. At the same time, a very
complicated distribution of $T(1.627\ \mu m)$ for CER < ~7 μm, as discussed above, can be an important factor for errors
noted for relatively small CERs. Further, the appearance of same $T(\lambda)$ values for larger CODs, as discussed above, can
be the next important factor for errors noted within COD < ~2.

210       Overall speaking, the retrieval error in COD is smaller than that for CER in terms of domain coverage and error

magnitude, suggesting that the transmittance based cloud remote sensing can have better effectiveness for COD
retrieval than for CER retrieval. Except for a limited number of error domains discussed above, the retrieval errors are
small in magnitude. For example, for COD > 15 and all types of errors, the 5, 50, and 95 percentile error values for
retrieved COD are -2.0%, -0.6% and 0.82%, respectively; they are -4.1%, -0.51% and 7.2%, respectively for retrieved
CER. For reference, the maximum (minimum) retrieval errors for COD ≥ 20 and CER = 10 μm for a spectral method
proposed by McBride et al. (2011) are ~7% (~2%) and ~52% (~14%) for COD and CER, respectively. Below, section 5
further sheds light on the quality of retrieved cloud properties based on compassion with standard data obtained from
independent sources.

**5    Comparison with data from independent sources**
**5.1 Solar radiation data**
The broad-band radiance and irradiance of shortwave spectral range (0.3 – 2.8 μm) observed at Chiba
(35.62°N,140.10°E) at each 20 seconds from December 2015 to December 2016 using a narrow-angle (NA) radiometer
(FOV: 5°) and a pyranometer (FOV: 180°), respectively are used for the evaluation of sky radiometer observed cloud
properties. The cloud properties from the sky radiometer are combined with MODIS observed surface albedo and
radiosonde observed PWC to calculate the counterparts of observation. A comparison is done for an average of 5
minutes observation of solar radiation that centers the sky radiometer observation time. Figures 5(a) and 5(b) show the
comparison for broad-band radiance and irradiance, respectively. For reference, such comparison is done also for
modeled values using cloud properties of AHI instead of sky radiometer. They are shown in Figs. 5(c) and 5(d) for
broad-band radiance and irradiance, respectively.

231       Firstly, one can see a strong (weak) correlation between modeled and observed values for broad-band radiance

(irradiance) when cloud properties from sky radiometer are used. One the other hand, the correlation between modeled
and observed values for broad-band radiance (irradiance) are weak (strong) for AHI cloud properties. Rather than
pyranometer, the NA radiometer observed data can best describe the quality of sky radiometer cloud properties because
of narrow FOV. A very good agreement noted in Fig. 5(a) with correlation coefficient ($r$) as strong as 0.93 suggests that
sky radiometer based cloud properties (at least COD) are highly qualitative. As the contribution of COD is dominant
over CER in broad-band solar radiation (Khatri et al., 2018), Fig. 5(a) alone cannot justify the quality of retrieved CER.
At the same time, the relatively poor agreement for irradiance comparison shown in Fig. 5(b) can be described due to
significantly different FOV of sky radiometer and pyranometer. It is because the surface observed solar radiation can be
drastically different depending on the instrument FOV. As an example, Fig. 6 shows the scatter plot between
broad-band irradiance and radiance observed by pyranometer and NA radiometer, respectively at Chiba during January
– March , 2016. The correlation between them is very poor. One of the important factors that deteriorates the correlation
between them is the cloud horizontal inhomogeneity. This can plausibly explain the poor agreement in Fig. 5(b) despite
reasonably accurate retrieval from the sky radiometer as evidenced by Fig. 5(a). On the contrary, AHI cloud properties
are average (or representative) values of specific coverage, i.e., pixel (e.g., 1km x 1km). As a result, irradiances
modeled from AHI cloud properties become closer with observed irradiance than those modeled from sky radiometer
cloud properties. It is because the sky radiometer observed cloud can be just a small portion of a pixel containing
horizontally inhomogeneous clouds.

**5.2 Satellite cloud products**
As part of validating water cloud products of MODIS and AHI using surface radiation data, Khatri et al. (2018)
compared water cloud properties retrieved from sky radiometer with those of MODIS and AHI observations for three
observation sites of SKYNET: Chiba, Hedo-misaki, and Fukue-jima for the period of October, 2016 to December, 2017.
They further used surface irradiance data, and found that the validation results using sky radiometer and surface
irradiance data are qualitatively same. A fairly good (poor) agreement was shown for COD (CER) between sky
radiometer and satellite products in Khatri et al. (2018) . They compared sky radiometer results with results of
collocated satellite pixels by selecting samples with time difference less than 1.25 minutes, which is half of temporal
resolution of AHI observation over Japan, and the distance between the pixel center and the observation site less than 1
km, and further doing parallax correction for satellite products
It is learnt from section 5.1 that the inhomogeneous clouds and/or broken clouds contained within the satellite
pixels are major obstacles in quality assessment of satellite products using sky radiometer results and vice versa. This
section attempts to improve our understating regarding the quality of sky radiometer products by using satellite products
effectively. For this purpose, we prepare samples for the comparison by addressing the cloud inhomogeneity related
problem in a logical way with available information at hand. If surface irradiance calculated from sky radiometer cloud
properties agree well with that observed at surface, the effective COD of actual inhomogeneous clouds may be
represented by a sky radiometer COD. Here, effective COD refers to COD of assumed plane-parallel homogenous
cloud layers which can produce irradiance equivalent to that produced by actual inhomogeneous clouds, i.e., measured
irradiance. Note that the satellite cloud properties retrieved from reflected signals assume clouds as plane-parallel
homogenous layers. The sky radiometer cloud properties that generate surface irradiances equivalent to observed values
by differing not more than 1% are selected to compare with satellite cloud properties. Figures 6(a) and 6(b) show the
comparison of sky radiometer CODs with MODIS and AHI values, respectively for sites and period same to Khatri et al.
(2018). The COD agreement is fairly good. The results are qualitatively same for both MODIS and AHI, by showing *r*
values of ~0.6 and ~0.7 and RMSE values of ~13 and ~10 for MODIS and AHI, respectively. Despite several


differences between sky radiometer and satellite products from both observation and retrieval perspectives, a fairly
good agreement indicates that they can have similar response towards thin and thick clouds, though the COD value may
not be exactly same. Similarly, Figs. 7(a) and 7(b) show the comparison of sky radiometer CERs with MODIS and AHI
values, respectively. The water absorbing wavelengths corresponding to MODIS and AHI are 2.1 μm and 3.79 μm,
respectively. The CERs between sky radiometer and satellite sensors are poorly correlated. One can see $r$ less than 0.12
and RMSE of ~7 μm for both satellite sensors. Such a poor correlation can be mainly due to the fact that satellite
sensors using reflected signals are highly sensitive towards cloud top layers (Platnick, 2000), whereas the sky
radiometer is sensitive to whole cloud layers.

**6    Conclusions**
In an effort of making cloud observation from surface more common and convenient, this study develops an algorithm
to retrieve cloud properties (COD and CER) from spectral zenith radiances measured by sky radiometer. By taking into
account of a priori information of a state vector and errors related to observation based transmittance and used ancillary
data (PWC and surface albedo), an optimal-estimation approach is proposed by fitting observation based transmittances
at the wavelengths of 0.87, 1.02, and 1.627μm with modeled values. To further ease data analysis of long-term
observation by overcoming the filter degradation problem, an on-site method of calibration for direct intensity is
proposed by making use of aerosol data of very clear sky days. The calibration constants derived from the proposed
method agree quite well with values determined by collocating the field instruments with the master instrument. The
retrieval error analyses performed by considering known ranges of errors in observation based transmittances and
ancillary data suggest a good performance of the algorithm, except for a certain narrow bands of small COD and CER
values. In general, the errors in input information can affect CER retrieval more significantly than COD retrieval, and
the retrieved CER can have relatively large errors when clouds are optically thin (COD < ~15) and cloud droplets are
small (CER < ~7 μm) or large ( CER > ~13 μm) in size. As part of quality assessment, cloud properties retrieved from
the proposed algorithm are compared indirectly with surface observed radiance/irradiance data and directly with
MODIS and AHI observed cloud properties. The retrieved cloud properties are found to produce the broadband
shortwave radiances quite similar to those observed by a narrow-angle radiometer, confirming the good quality of
retrieved products (at least COD) from sky radiometer. However, the agreement is relatively poor when broadband
shortwave irradiances observed by a pyranometer of wide FOV are compared with the modeled values. It is likely due
to distinctly different FOVs of sky radiometer and pyranometer, suggesting a very important influence of instrument's
FOV on cloud remote sensing. Further, a fairly good agreement of COD between sky radiometer and satellite sensors
can be seen; however, the agreement is poor for CER comparison.

*Code/Data Availability:* Data and retrieval code are available from the corresponding author upon request.



*Author Contribution*: PK, HI, and TH developed study framework and code. HI, TT, AY, and AD generated data. HL
and QK helped in advancing study framework and manuscript writing. All co-authors read the manuscript and provided
suggestions and comments.

*Competing interests:* The authors declare that they have no conflict of interest.

*Acknowledgements:* This research is supported by the 2nd Research Announcement on the Earth Observations of the
Japan Aerospace Exploration Agency (JAXA) (PI No. ER2GCF211, Contract No. 19RT000370), a Grant-in-Aid for
Scientific Research (C) 17K05650 from Japan Society for the Promotion of Science (JSPS), "Virtual Laboratory for
Diagnosing the Earth's Climate System" program of MEXT, Japan, and CREST/JST research fund of grant number
JPMJCR15K4.

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


**Figures**

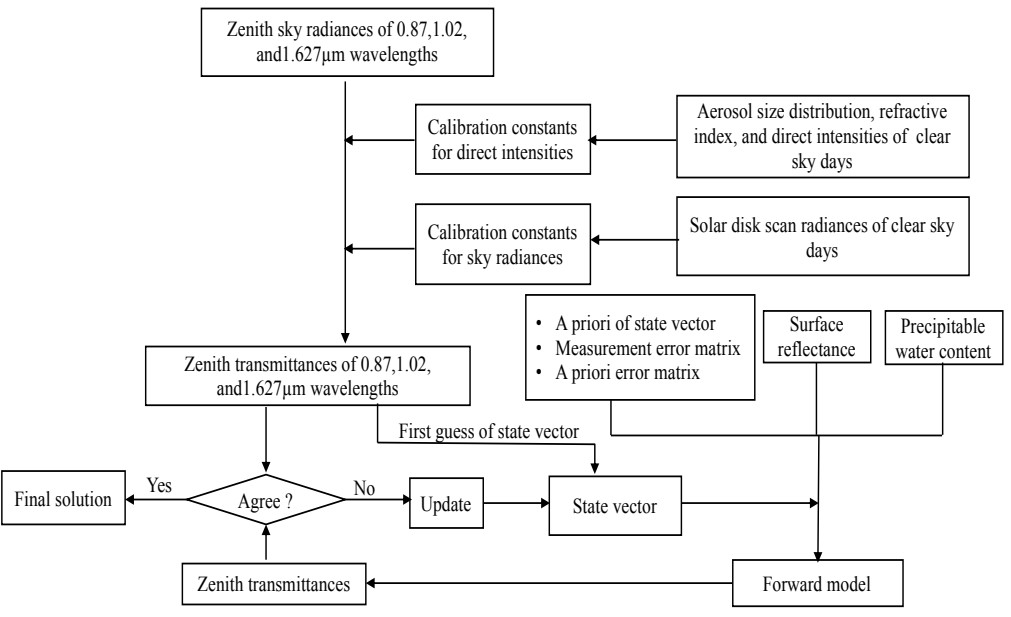


**Figure 1: A schematic diagram of study method.**












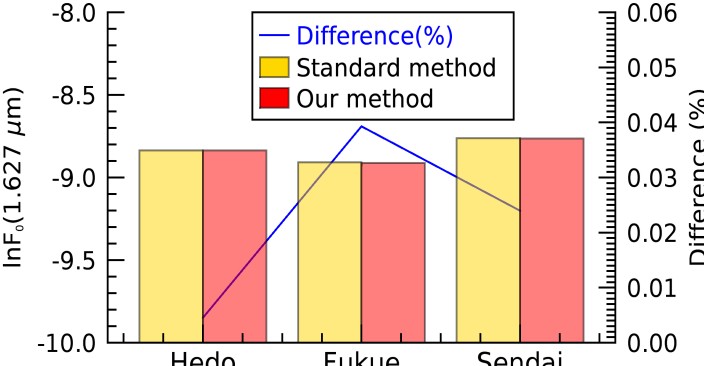


**Figure 2: Comparison of direct intensity calibration constant (lnF$_0$) values at water absorbing wavelength of 1.627 μm between the standard method (calibration using the master instrument) and an on-site method proposed in this study for sky radiometers belonging to Hedo-misaki (26.87°N, 128.25°E), Fukue-jima (32.75°N, 128.68°E), and Sendai (38.26°N,140.84°E). Shown in the figure is also the difference (%), i.e., the difference (in percentage) between proposed method and the standard method normalized by the value of the standard method.**

437

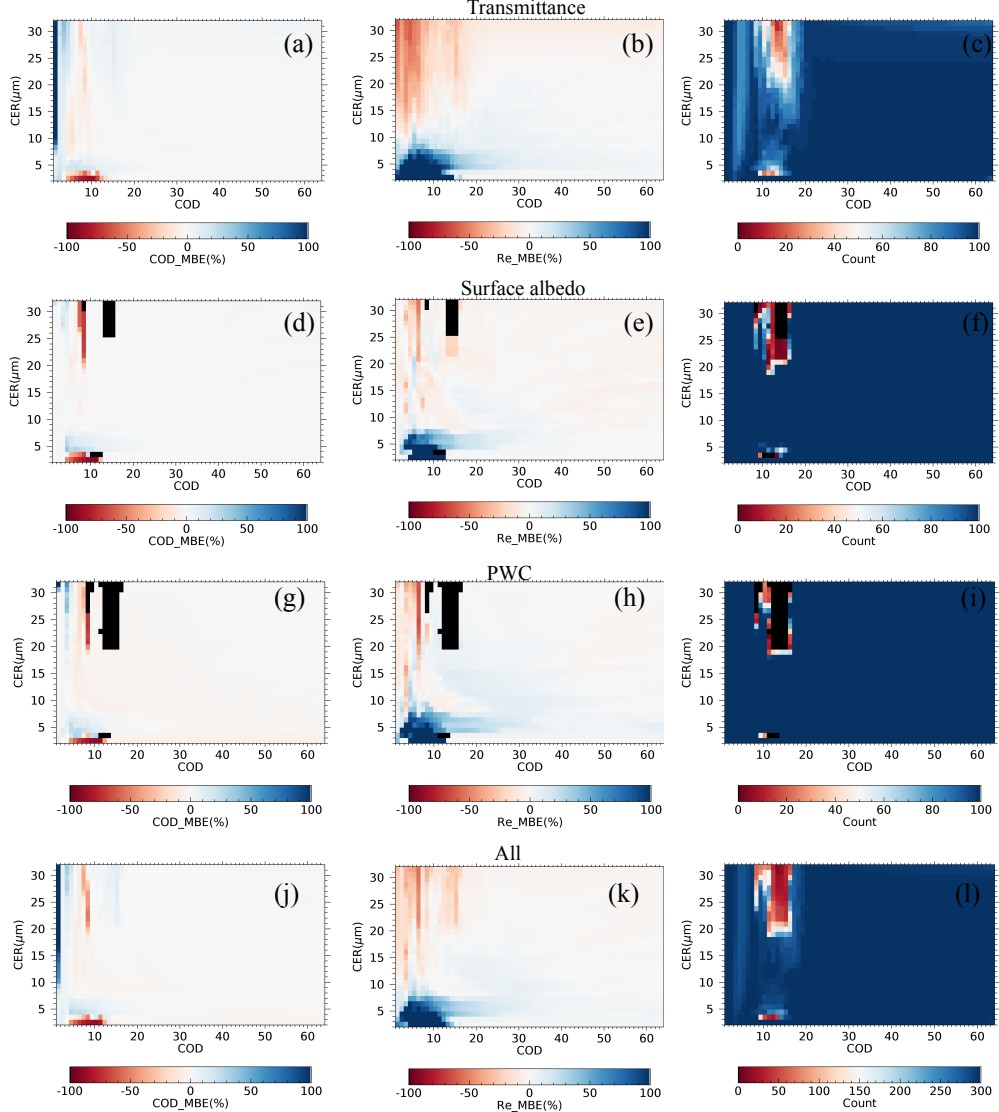

438

**Figure 3: Mean bias error (MBE) values for retrieved COD and CER and total number of successful retrieval in**

**(a), (b) and (c), respectively for assumed error in transmittance; (d) – (f): same as upper panel but for assumed**

**error in surface albedo; (g) – (i): same as upper panel but for assumed error in precipitable water content; (j) –**

**(l): same as upper panel but for all error sources. The 100% unsuccessful retrieval is denoted by black color.**





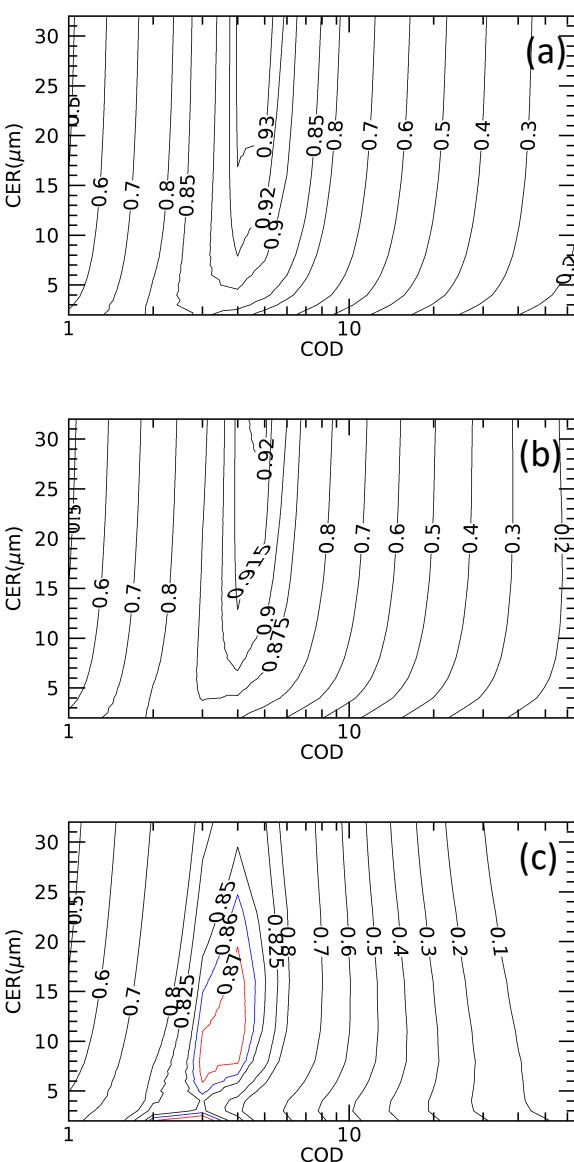


**Figure 4: Contour plots of transmittances at the wavelengths of (a) 0.87 µm, (b) 1.02 µm, and 1.627 µm for solar**

**zenith and azimuth angles of 30° and 0°, respectively. The values of transmittances are given within the contour**

**lines. Different colors are used for 1.627 µm to make it easy to distinguish.**









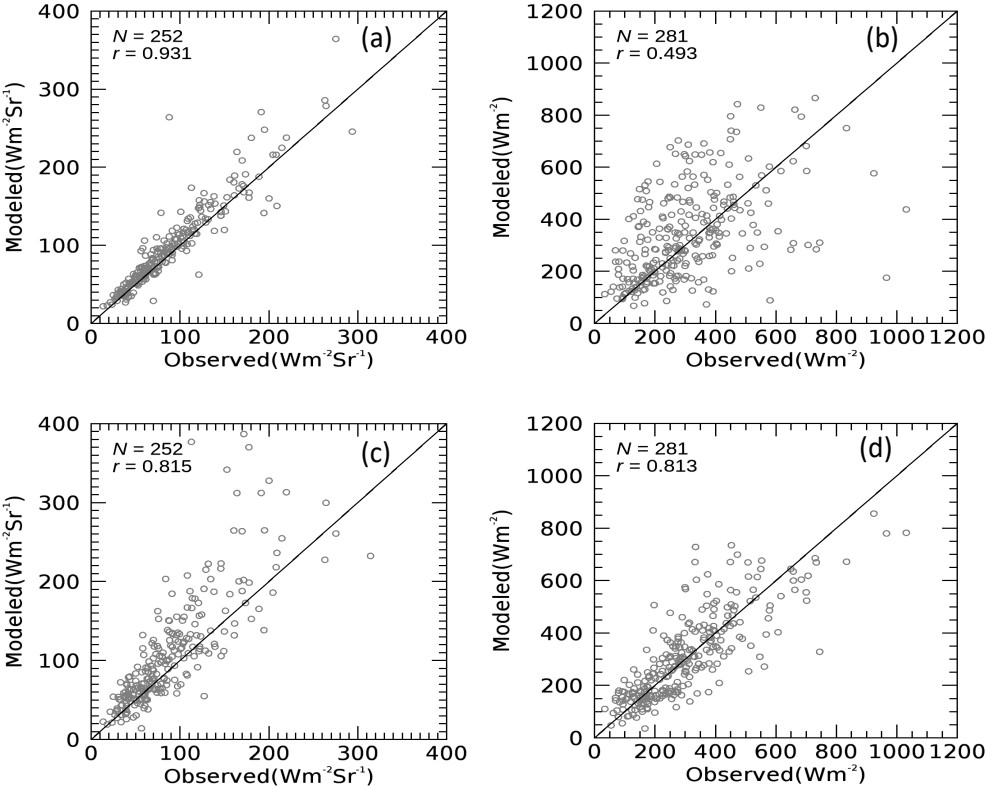


**Figure 5: Comparison between modeled and observed broad-band (a) radiances and (b) irradiances for modeled**


**values using sky radiometer cloud proprieties for observation site of Chiba (35.62°N,140.10°E) for 2016.**


**Similarly, (c) and (d) show the comparison results for broad-band radiances and irradiance, respectively for**


**modeled cloud properties corresponding to AHI.**














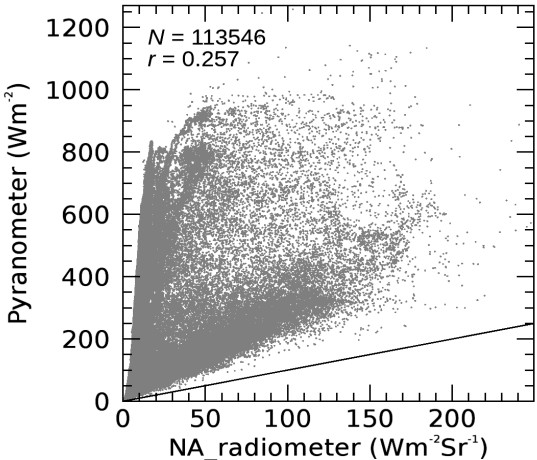


**Figure 6: Scatterplot of broad-band radiances and irradiances observed by radiometers of narrow-angle and wide-angle (pyranometer), respectively at Chiba (35.62°N,140.10°E) during January – March, 2016.**






















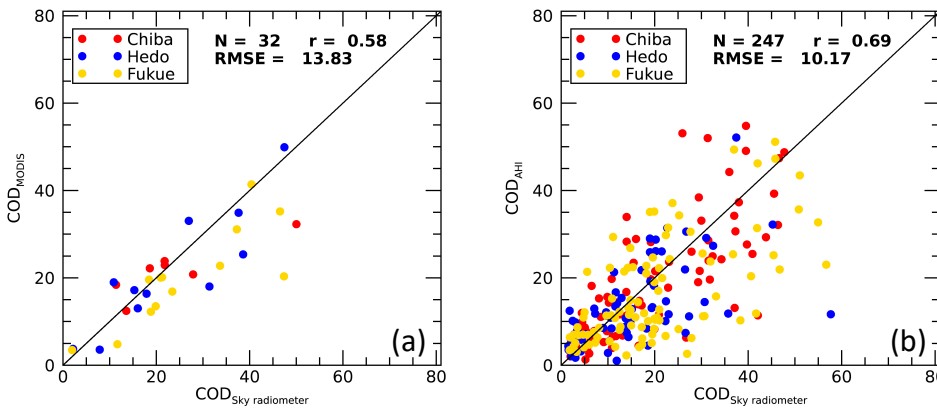


**Figure 7: Comparison of sky radiometer CODs with (a) MODIS and (b) AHI CODs for observation sites of Chiba (35.62°N,140.10°E), Hedo-misaki (26.87°N, 128.25°E), Fukue-jima (32.75°N, 128.68°E) for periods of October, 2015 to December, 2016.**














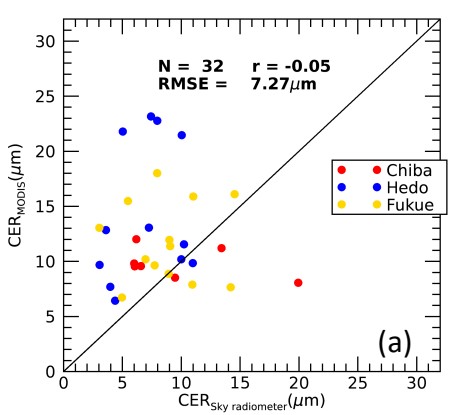
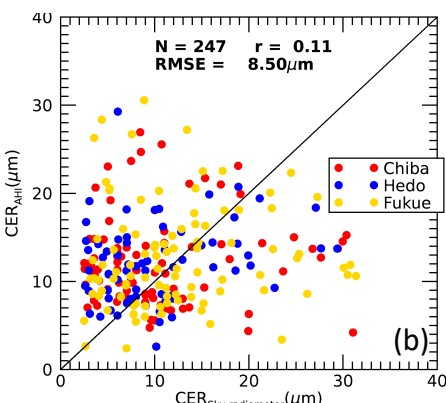

**Figure 8: Same as Figure 7, but for CER comparison.**

