# Peer review of "Retrieval of cloud properties from spectral zenith radiances observed by sky radiometers"

_Atmospheric Measurement Techniques, 2019_

## Referee Comment (RC1) · Anonymous Referee #1 · 30 Jul 2019

The article entitled Retrieval of cloud properties from sky radiometer observed spectral zenith radiances, by P. Khatri, H. Iwabuchi, T. Hayasaka, H. Irie, T. Takamura, A. Yamazaki, A. Damiani, H. Letu and Q. Kai presents and validates a method for retrieving cloud properties (i.e. cloud optical depth and Cloud effective radius) from zenith radiance measurements of the sky, performed from ground with a commercial available sun-sky radiometer Prede POM02.

The article is considered important and presents a method that means a step forward in the establishment and improvement of ground based methods for the observation of cloud properties, that are one of the most important factors in the Earth climate. The establishment, validation, and further improvement of the method will allow other users of the Prede POM02 sun-sky radiometer, particularly users of the SKYNET international network, to obtain relevant information and contribute to this field of research.

The study is considered adequate for this journal. However, the article would benefit of a final revision by a native English speaker.

Specific comments: - Abstract, line 17: the procedure of deriving calibration constants from another instrument could be called "calibration transfer method". - Line 34-35: please rewrite sentence ("unlike advancements"?) - Line 47: not sure "tally" is the most appropiate word here - Line 80: you have included 940 nm channel in the list of channels used for aerosol retrieval? - Line 89 and 91: instead of "alternation", "alternative" could suit best - Line 92: rewrite "A more detailed study about..." - Line 96-98: authors state that 2.2um channel is not used because the longest wavelength used by AERONET is 1.627um, but Cimels are not used in this study. Do the authors plan to apply the method on Cimel instruments in the future? Otherwise, I think it is not well understood the reason for rejecting this channel. - Line 103: "from the solar disk scan during very clear sky days" - Line 105: "wavelengths" - Line 111: "very mere" -> very small - Line 113: do the authors expect any limitation in the method for cases of dust mixed with clouds due to non-sphericity? - Line 116: Beer-Lambert - Line 120: do the authors use any minimum number of data points to perform a succesful final IL fit? Any other threshold or criteria? - Line 129: "temporal variation of lnF0" - Line 154-155: "calculated for COD and CER in the intervals 1-64 and 2-32 respectively, with steps of 1" - Line 160: the assumed error of 1.0 cm for PWC looks like a very high upper estimation of error. Is it a typo error? - Lines 169-203: it is a long paragraph. Perhaps it could be divided at lines 183 and 191 - Lines 213-214: do the three percentages correspond respectively to percentiles 5, 50 and 95? - Line 217: comparison, not compassion - Line 221 (section 5.1). Is the NA radiometer a pirheliometer? Is it pointing at nadir direction continuously? Does it really measure radiance, or irradiance? I think it would useful to have some more details about the instrumentation used here. - Line 236: "highly qualitative"? - Section 5.2.: in order to better understand the improvement of the comparison respect to Kathri 2018, a short mention to the previous results using

all the database would be useful

Figures: - Line 270: Figures 7a and 7b - Line 276: Figures 8a and 8b - Figure 4: I understand that figures a, b, c correspond to channels 870, 1020 and 1627. But I do not understand what are zenith and azimuth angles respectively. If both zenith and azimuth results are the same for figures a and b, but are slightly different for figure c, please state that zenith and azimuth angles are represented with different colors.

---

## Referee Comment (RC2) · Anonymous Referee #2 · 21 Aug 2019

General comments:

The development of the algorithm in this study is important because it is of great merit for radiometers to make their self calibrations on-site. However, there are some issues to be clarified before publication finally, in terms of the detailed points below. In addition, it is recommended to have the manuscript English-proofed.

Specific comments:

L67-68 "We use three carefully selected wavelengths to retrieve COD and CER simultaneously." How did the authors carefully select the wavelengths?

L95 Fig. 1 Do the authors specify some criterion for the number of iteration? Do all the observed data properly retrieved?

[Figure]

L181 Fig.4 From Fig.4, it seems that transmittances of the three wavelengths have dual values for a certain effective radius of cloud droplet. For example, the values 0.7 of transmittance appear at two regions of cloud optical depth more than and less than 10. The situation might cause the problem of dependency on a-priori or a starting value of the iteration in Fig. 1 and x_a in Eq. 2. Does the issue is not critical for this study?

L199 "Note that the absorption tends to reduce $T(\lambda)$, whereas the forward scattering tends to increase it." In terms of cloud retrieval, multiple scattering is important for larger COD, which will enhance the forward scattering and absorption processes.

L223-224 "a narrow-angle (NA) radiometer (FOV: 5°)" The observation was conducted in the zenith direction same as the sky radiometer?

Fig. 6 The slope of the solid line should be multiplied by 2*phi. Currently, the pyranometer observation seems overestimated greatly, compared with narrow angle radiometer, which might mislead to inconsistency of the two observations.

Technical corrections:

L28 "Foster" -> "Forster"

L59 "MCBride" -> "McBride"

L120 "(r)" rather than "(r^2)"

L163 "priory" -> "a priori"

L169 Fig. 3b, 3e, 3h, and 3k The style of right axis should be same as the other panels.

L236 "qualitative" -> "quantitative" ?!

L270 "Figures 6(a) and 6(b)" -> "Figures 7a and 7b"

Figure 7, Legends: Please omit the dual markers for each site.

L276 "Figs. 7(a) and 7(b)" -> "Figs. 8a and 8b"

---

## Author Comment (AC1) · 26 Sep 2019

**Response to comments of Reviewer 1**

Authors would like to express sincere thanks to an anonymous reviewer for his/her valuable comments. We revised a manuscript very carefully based on given comments. The comments of the reviewer are in blue, our replies are in black, and changes made in the revised manuscript are in red. Our replies to the comments are as below.

The article entitled Retrieval of cloud properties from sky radiometer observed spectral zenith radiances, by P. Khatri, H. Iwabuchi, T. Hayasaka, H. Irie, T. Takamura, A. Yamazaki, A. Damiani, H. Letu and Q. Kai presents and validates a method for retrieving cloud properties (i.e. cloud optical depth and Cloud effective radius) from zenith radiance measurements of the sky, performed from ground with a commercial available sun-sky radiometer Prede POM02.

The article is considered important and presents a method that means a step forward in the establishment and improvement of ground based methods for the observation of cloud properties, that are one of the most important factors in the Earth climate. The establishment, validation, and further improvement of the method will allow other users of the Prede POM02 sun-sky radiometer, particularly users of the SKYNET international network, to obtain relevant information and contribute to this field of research. The study is considered adequate for this journal. However, the article would benefit of a final revision by a native English speaker.

➔ Thank you for your encouraging comments. The revised manuscript has been read by a native English speaker.

Specific comments:

- Abstract, line 17: the procedure of deriving calibration constants from another instrument could be called "calibration transfer method".

➔ It is done in the revised manuscript.

- Line 34-35: please rewrite sentence ("unlike advancements"?)

➔ Those sentences are rewritten as below.

Compared with the routine observation of aerosols through surface networks, such as AERONET (https://aeronet.gsfc.nasa.gov/) and SKYNET (http://atmos3.cr.chiba-u.jp/skynet/), observation of clouds from the surface is performed at a limited number of stations and most of the observation data are not easily accessible.

- Line 47: not sure "tally" is the most appropriate word here

➔ We rewrote the sentence as below.

The fundamental idea is to compare the observed signals with LUT data corresponding to prior known cloud optical depth (COD) and cloud particle effective radius (CER) while finding a plausible solution for the COD and CER combination.

- Line 80: you have included 940 nm channel in the list of channels used for aerosol retrieval?

➔ It is corrected in the revised manuscript.

- Line 89 and 91: instead of "alternation", "alternative" could suit best

➔ It is done in the revised manuscript.

- Line 92: rewrite "A more detailed study about..."

➔The typo mistake has been corrected. In the revised manuscript, the sentence appears as

 A more detailed study about sky radiometers and their calibrations can be found in Khatri et al. (2016).

- Line 96-98: authors state that 2.2um channel is not used because the longest wavelength used by AERONET is 1.627um, but Cimels are not used in this study. Do the authors plan to apply the method on Cimel instruments in the future? Otherwise, I think it is not well understood the reason for rejecting this channel.

➔We clarified the reasons in the revised manuscript as below.

We use sky radiances ($E$) observed at three longer wavelengths (0.87, 1.02, and 1.627 μm), excluding 2.2 μm, which is not used for two main reasons. First, our statistical analysis suggests that the number of unphysical data (observation data recorded as 0) for 2.2 μm is high; thus, 2.2 μm is excluded to increase the retrieval number. Second, the longest wavelength used by AERONET is 1.64 μm; so the proposed algorithm could be easily used for sun photometer observed data as well. Wavelengths shorter than 0.87 μm are not used to avoid the effect of aerosols as far as possible.

 - Line 103: "from the solar disk scan during very clear sky days"

➔ The sentence is corrected as suggested by the reviewer as

$\Delta\Omega$ for 0.87, 1.02, and 1.627 μm can be determined from the solar disk scan during very clear sky days (Nakajima et al., 1996)

- Line 105: "wavelengths"

➔ The typo mistake is corrected in the revised manuscript.

- Line 111: "very mere" -> very small

➔ A correction is made as suggested by the reviewer in the revised manuscript.

–Line 113: do the authors expect any limitation in the method for cases of dust mixed with clouds due to non-sphericity?

➔We used aerosol data of very clear sky days while determining the calibration constant ($F_0$) values using the proposed method. Thus, there is a very little chance of mixing clouds with aerosols as suggested by the reviewer. However, even if some cloudy data have been misinterpreted as aerosols by a cloud-screening procedure, such cloudy data can get filtered because the proposed method eliminates the outlier that decreases the correlation coefficient between $lnF$ and $\tau m$ ($F$, $\tau$, and $m$ are measured direct intensity, total optical thickness and optical air mass, respectively) through an iteration process until a very strong correlation ($r \geq$ 0.997) is obtained.

–Line 116: Beer-Lambert

➔ The typo mistake is corrected in the revised manuscript.

- Line 120: do the authors use any minimum number of data points to perform a successful final IL fit? Any other threshold or criteria?

➔ There is no specific condition regarding minimum number of data points because each final IL plot is visually inspected to confirm that suspicious data points are not included and there are enough number of data points in the fit.

The proposed method was designed by putting other criteria, such as the maximum value of aerosol optical thickness (AOT) and solar zenith angle, as well. However, our test indicates that such criteria can have a very less influence in overall performance because the outlier is removed through iteration as explained in the reply of the previous comment. Thus, criteria other than the value of correlation coefficient are removed.

➜ The sentence is corrected as below.

Thus, the proposed method can be used to determine temporal variation of $lnF_{0,\,1.627}$, which is useful for analyzing long-term observation data by mitigating the filter degradation problem.

- Line 154-155: "calculated for COD and CER in the intervals 1-64 and 2-32 respectively, with steps of 1"

➜The sentence is corrected as below.

The retrieval errors are calculated for COD and CER values in the ranges of 1–64 and 2–32 μm, respectively, in steps of 1 μm.

- Line 160: the assumed error of 1.0 cm for PWC looks like a very high upper estimation of error. Is it a typo error?

➜ There was no typo mistake. We assumed error value of this magnitude to better understand the algorithm performance.

- Lines 169-203: it is a long paragraph. Perhaps it could be divided at lines 183 and 191

➜ It is done in the revised manuscript.

- Lines 213-214: do the three percentages correspond respectively to percentiles 5, 50 and 95?

➜ Yes, they are according to the reviewer. The English writing of the revised manuscript has been polished by a native speaker. Thus, each sentence is expected to have a clear meaning in the revised manuscript.

➜ This typo mistake is corrected in the revised manuscript.

- Line 221 (section 5.1). Is the NA radiometer a pirheliometer? Is it pointing at nadir direction continuously? Does it really measure radiance, or irradiance? I think it would useful to have some more details about the instrumentation used here.

➜ The description is elaborated in the revised manuscript as below.

The broad-band radiance and irradiance of the shortwave spectral range (0.3 – 2.8 μm) observed using a narrow-angle radiometer (EKO Instruments Co., Ltd., Japan; FOV: 5°) and a

pyranometer (Kipp and Zonen, Netherlands; FOV: 180°), respectively, at Chiba (35.62°N, 140.10°E) every 20 s from December 2015 to December 2016 are used to evaluate the cloud properties observed by the sky radiometer. The narrow-angle radiometer observes the downwelling irradiance signals as voltage in a narrow FOV. The instrument was calibrated by the manufacturer in the laboratory, and the observed signals are converted into radiance (unit: W/m$^2$/sr) by using the company provided calibration constant value. Because the narrow-angle radiometer faces upward, thus obtained radiance is from the zenith.

- Line 236: "highly qualitative"?

➔ "qualitative enough" is used instead of "highly qualitative" in the revised manuscript.

- Section 5.2.: in order to better understand the improvement of the comparison respect to Kathri 2018, a short mention to the previous results using all the database would be useful

➔As suggested by the reviewer, a new paragraph is added in section 5.2.

Although the qualitive information reported by Khatri et al. (2018) and the comparisons in Figures 7 and 8 of this study are similar, there are differences in Figures 7 and 8 with the comparison plots shown in Khatri et al. (2018). The application of data screening criteria in this study generally screened out data with large differences between the sky radiometer and satellite sensors. These large differences in the previous comparison probably arose from the different FOVs of the satellite sensor and sky radiometer, while observing inhomogeneous clouds. Thus, the comparison results presented in this study by addressing the cloud inhomogeneity problem more logically should give more accurate and refined information than those presented in Khatri et al. (2018).

Figures: - Line 270: Figures 7a and 7b

➔ The typo mistake is corrected in the revised manuscript.

- Line 276: Figures 8a and 8b

➔ The typo mistake is corrected in the revised manuscript.

- Figure 4: I understand that figures a, b, c correspond to channels 870, 1020 and 1627. But I do not understand what are zenith and azimuth angles respectively. If both zenith and azimuth results are the same for figures a and b, but are slightly different for figure c, please state that zenith and azimuth angles are represented with different colors.

➔The zenith and azimuth angles are exactly same for Figs. 4a, 4b, and 4c. Since the transmittance changes with the solar position, we choose solar zenith and azimuth angles of 30° and 0° as representative to investigate how the transmittances of 0.87, 1.02, and 1.627 μm

change with the changes of COD and CER. Therefore, the solar zenith and azimuth angles are indicated in the caption.

If Fig. 4c is viewed very carefully, the contour lines denoted by different colors for 2 < COD < 5 and CER > 4 μm appear again for ~2 < COD < 4 and CER < 4 μm. It is technically difficult (due to not sufficient space) to write the values of transmittances within the contour lines (similar to Figs. 4a and 4b) in the second domain, i.e., ~2 < COD < 4 and CER < 4 μm. Therefore, different colors are used to resolve this technical difficulty.

**References**

Nakajima, T., Tonna, G., Rao, R., Kaufman, Y., and Holben, B. N. : Use of sky brightness measurements from ground for remote sensing of particulate polydispersions, Appl. Opt., 35(15), 2672–2686, https://doi.org/10.1364/AO.35.002672, 1996.

Khatri, P., Takamura, T., Estellés, V., Irie, H., Kuze, H., Campanelli, M., Sinyuk, A., Lee, S.-M., Sohn, B. J., Pandithurai, G., Kim, S.-W., Yoon, S. C., Martinez-Lozano, J. A., Hashimoto, M., Devara, P. C. S., and Manago, N.: Factors for inconsistent aerosol single scattering albedo between SKYNET and AERONET, J. Geophys. Res., 121(4), 1859-1877, https:/doi.org/10.1002/20159JD023976, 2016.

Khatri, P., Hayasaka, T., Iwabuchi, H., Takamura, T., Irie, H., and Nakajima, T. Y.: Validation of MODIS and AHI observed water cloud properties using surface radiation data, J. Meteor. Soc. Japan, 96B, 151-172, https:/doi.org/10.2151/jmsj.2018-036, 2018.

---

## Author Comment (AC2) · 26 Sep 2019

**Response to comments of Reviewer 2**

Authors would like to express sincere thanks to an anonymous reviewer for his/her valuable comments. We revised a manuscript very carefully based on given comments. The comments of the reviewer are in blue, our replies are in black, and changes made in the revised manuscript are in red. Our replies to the comments are as below.

The development of the algorithm in this study is important because it is of great merit for radiometers to make their self-calibrations on-site. However, there are some issues to be clarified before publication finally, in terms of the detailed points below. In addition, it is recommended to have the manuscript English-proofed.

➔ Thank you for your encouraging comment. The manuscript has been read by a professional native English speaker.

Specific comments:

L67-68 "We use three carefully selected wavelengths to retrieve COD and CER simultaneously." How did the authors carefully select the wavelengths?

➔ They are described in the revised manuscript as below.

We use sky radiances ($E$) observed at three longer wavelengths (0.87, 1.02, and 1.627 μm), excluding 2.2 μm, which is not used for two main reasons. First, our statistical analysis suggests that the number of unphysical data (observation data recorded as 0) for 2.2 μm is high; thus, 2.2 μm is excluded to increase the retrieval number. Second, the longest wavelength used by AERONET is 1.64 μm; so the proposed algorithm could be easily used for sun photometer observed data as well. Wavelengths shorter than 0.87 μm are not used to avoid the effect of aerosols as far as possible.

L95 Fig. 1 Do the authors specify some criterion for the number of iteration? Do all the observed data properly retrieved?

➔ The total number of iterations is set as 50. If the solution does not converge within 50 iterations, the analysis is discarded.

L181 Fig.4 From Fig.4, it seems that transmittances of the three wavelengths have dual values for a certain effective radius of cloud droplet. For example, the values 0.7 of transmittance

appear at two regions of cloud optical depth more than and less than 10. The situation might cause the problem of dependency on a-priori or a starting value of the iteration in Fig. 1 and x_a in Eq. 2. Does the issue is not critical for this study?

➔The *a-priori* values are climatological data sets, and they are fixed in the algorithm. The starting values of COD and CER can have an important effect in the retrieval as suggested by the reviewer. Our approach to overcome this problem is described in the revised manuscript as below.

As highlighted in Sections 1 and 4, transmittance signals may not always be characterized by unique COD or CER values. Consequently, the initial values of COD and CER used for iteration can be important when searching the plausible set of COD and CER values. To address this important issue, we first approximate the initial COD and CER values to start the iteration. The approximation is done by searching a set of COD and CER values by comparing observed $T_{1.627}/T_{1.02}$ and $T_{1.02}$ with LUT of corresponding values modeled for COD values of 1–64 and CER values of 2–32 µm in steps of 1 µm. $T_{1.627}/T_{1.02}$ generally decreases with the increase of COD; whereas when COD increases, $T_{1.02}$ increases first until reaching the peak value, and then starts to decrease. Thus, $T_{1.627}/T_{1.02}$ and $T_{1.02}$ can be used simultaneously to determine the range of COD and CER in which the true values are likely to fall. A set of COD and CER values that generate the smallest root mean square difference between the observed and modeled values is used for the initial values in the iteration.

L199 "Note that the absorption tends to reduce T( ), whereas the forward scattering tends to increase it." In terms of cloud retrieval, multiple scattering is important for larger COD, which will enhance the forward scattering and absorption processes.

➔We refined the sentences as suggested by the reviewer as

Both the forward scattering and absorption can increase with the increase of COD along with the increase in multiple scattering; the increase in $T(\lambda)$ before the peak value is due to the dominance of forward scattering over absorption, and vice versa for the decrease in $T(\lambda)$ after the peak value.

L223-224 "a narrow-angle (NA) radiometer (FOV: 5)" The observation was conducted in the zenith direction same as the sky radiometer?

➔ We provided information below to clarify this issue.

The instrument was calibrated by the manufacturer in the laboratory, and the observed signals are converted into radiance (unit: $W/m^2/sr$) by using the company provided calibration constant

value. Because the narrow-angle radiometer faces upward, thus obtained radiance is from the zenith.

Fig. 6 The slope of the solid line should be multiplied by 2*phi. Currently, the pyranometer observation seems overestimated greatly, compared with narrow angle radiometer, which might mislead to inconsistency of the two observations.

➔As suggested by the reviewer, the slope of the solid line is multiplied by 2*phi.

[Figure]

Figure 6: Scatterplot of broad-band radiances and irradiances observed with a narrow-angle radiometer and a wide-angle pyranometer at Chiba (35.62°N, 140.10°E) during January–March 2016. The solid line represents $y = 2\pi x$.

  The solid line hardly suggests the overestimation from pyranometer as data points are scattered in both sides of a solid line. On the other hand, it indicates the asymmetric distribution of radiance.

Technical corrections:
L28 "Foster" -> "Forster"
➔ It is corrected in the revised manuscript.

L59 "MCBride" -> "McBride"

➔ It is corrected in the revised manuscript.

L120 "(r)" rather than "(rˆ2)"

➔ It is done in the revised manuscript.

L163 "priory" -> "a priori"

➔ It is corrected in the revised manuscript.

L169 Fig. 3b, 3e, 3h, and 3k The style of right axis should be same as the other panels.

➔ They are done in the revised manuscript.

L236 "qualitative" -> "quantitative" ?!

➔ It is qualitative.

L270 "Figures 6(a) and 6(b)" -> "Figures 7a and 7b"

➔They are corrected in the revised manuscript.

Figure 7, Legends: Please omit the dual markers for each site.

➔They are done in the revised manuscript.

L276 "Figs. 7(a) and 7(b)" -> "Figs. 8a and 8b"

➔They are corrected in the revised manuscript.

---

## Author Response (AR1)

**Response to comments of Reviewer 1**

Authors would like to express sincere thanks to an anonymous reviewer for his/her valuable comments. We revised a manuscript very carefully based on given comments. The comments of the reviewer are in blue, our replies are in black, and changes made in the revised manuscript are in red. Our replies to the comments are as below.

The article entitled Retrieval of cloud properties from sky radiometer observed spectral zenith radiances, by P. Khatri, H. Iwabuchi, T. Hayasaka, H. Irie, T. Takamura, A. Yamazaki, A. Damiani, H. Letu and Q. Kai presents and validates a method for retrieving cloud properties (i.e. cloud optical depth and Cloud effective radius) from zenith radiance measurements of the sky, performed from ground with a commercial available sun-sky radiometer Prede POM02.

The article is considered important and presents a method that means a step forward in the establishment and improvement of ground based methods for the observation of cloud properties, that are one of the most important factors in the Earth climate. The establishment, validation, and further improvement of the method will allow other users of the Prede POM02 sun-sky radiometer, particularly users of the SKYNET international network, to obtain relevant information and contribute to this field of research. The study is considered adequate for this journal. However, the article would benefit of a final revision by a native English speaker.

→ Thank you for your encouraging comments. The revised manuscript has been read by a native English speaker.

**Specific comments:**

- Abstract, line 17: the procedure of deriving calibration constants from another instrument could be called "calibration transfer method".

 $\rightarrow$  It is done in the revised manuscript (Line 17).

**- Line 34-35: please rewrite sentence ("unlike advancements"?)**

→ Those sentences are rewritten as below (Line 32 – Line 35).

Compared with the routine observation of aerosols through surface networks, such as AERONET (https://aeronet.gsfc.nasa.gov/) and SKYNET (http://atmos3.cr.chiba-u.jp/skynet/), observation of clouds from the surface is performed at a limited number of stations and most of the observation data are not easily accessible.

- Line 47: not sure "tally" is the most appropriate word here

 $\rightarrow$  We rewrote the sentence as below (Line 44 – Line 46).

The fundamental idea is to compare the observed signals with LUT data corresponding to prior known cloud optical depth (COD) and cloud particle effective radius (CER) while finding a plausible solution for the COD and CER combination.

Line 80: you have included 940 nm channel in the list of channels used for aerosol retrieval?
→ It is corrected in the revised manuscript (Line 75 – Line 76).

- Line 89 and 91: instead of "alternation", "alternative" could suit best

 $\rightarrow$  It is done in the revised manuscript (Line 84 & Line 85).

- Line 92: rewrite "A more detailed study about..."

→ The typo mistake has been corrected. In the revised manuscript, the sentence appears as (Line 86 - Line 87).

A more detailed study about sky radiometers and their calibrations can be found in Khatri et al. (2016).

- Line 96-98: authors state that 2.2um channel is not used because the longest wavelength used by AERONET is 1.627um, but Cimels are not used in this study. Do the authors plan to apply the method on Cimel instruments in the future? Otherwise, I think it is not well understood the reason for rejecting this channel.

→ We clarified the reasons in the revised manuscript as below (Line 90 - Line 95).

We use sky radiances (*E*) observed at three longer wavelengths (0.87, 1.02, and 1.627  $\mu$ m), excluding 2.2  $\mu$ m, which is not used for two main reasons. First, our statistical analysis suggests that the number of unphysical data (observation data recorded as 0) for 2.2  $\mu$ m is high; thus, 2.2  $\mu$ m is excluded to increase the retrieval number. Second, the longest wavelength used by AERONET is 1.64  $\mu$ m; so the proposed algorithm could be easily used for sun photometer observed data as well. Wavelengths shorter than 0.87  $\mu$ m are not used to avoid the effect of aerosols as far as possible.

- Line 103: "from the solar disk scan during very clear sky days"

**→ The sentence is corrected as suggested by the reviewer as (Line 99 - Line 100)**

 $\Delta \Omega$  for 0.87, 1.02, and 1.627 µm can be determined from the solar disk scan during very clear sky days (Nakajima et al., 1996)

- Line 105: "wavelengths"

 $\rightarrow$  The typo mistake is corrected in the revised manuscript (Line 101).

- Line 111: "very mere" -> very small

 $\rightarrow$  A correction is made as suggested by the reviewer in the revised manuscript (Line 106).

-Line 113: do the authors expect any limitation in the method for cases of dust mixed with clouds due to non-sphericity?

→We used aerosol data of very clear sky days while determining the calibration constant ( $F_0$ ) values using the proposed method. Thus, there is a very little chance of mixing clouds with aerosols as suggested by the reviewer. However, even if some cloudy data have been misinterpreted as aerosols by a cloud-screening procedure, such cloudy data can get filtered because the proposed method eliminates the outlier that decreases the correlation coefficient between lnF and  $\tau m$  (F,  $\tau$ , and m are measured direct intensity, total optical thickness and optical air mass, respectively) through an iteration process until a very strong correlation ( $r \ge 0.997$ ) is obtained (Line 113 – Line 115).

**-Line 116: Beer-Lambert**

 $\rightarrow$  The typo mistake is corrected in the revised manuscript (Line 112).

- Line 120: do the authors use any minimum number of data points to perform a successful final IL fit? Any other threshold or criteria?

→ There is no specific condition regarding minimum number of data points because each final IL plot is visually inspected to confirm that suspicious data points are not included and there are enough number of data points in the fit.

The proposed method was designed by putting other criteria, such as the maximum value of aerosol optical thickness (AOT) and solar zenith angle, as well. However, our test indicates that such criteria can have a very less influence in overall performance because the outlier is removed through iteration as explained in the reply of the previous comment. Thus, criteria other than the value of correlation coefficient are removed (Line 113 – Line 115).

- Line 129: "temporal variation of lnF0"

→ The sentence is corrected as below (Line 123 – Line 124).

Thus, the proposed method can be used to determine temporal variation of  $lnF_{0,1.627}$ , which is useful for analyzing long-term observation data by mitigating the filter degradation problem.

- Line 154-155: "calculated for COD and CER in the intervals 1-64 and 2-32 respectively, with steps of 1"

→ The sentence is corrected as below (Line 158 – Line 159).

The retrieval errors are calculated for COD and CER values in the ranges of 1–64 and 2–32  $\mu$ m, respectively, in steps of 1  $\mu$ m.

- Line 160: the assumed error of 1.0 cm for PWC looks like a very high upper estimation of error. Is it a typo error?

→ There was no typo mistake for assumed error of PWC; however, there was a typo mistake for assumed value of PWC. The assumed value of PWC was 1.5 cm (it was mistakenly written as 1.0 cm). It is corrected in the revised manuscript. We assumed error value of this magnitude to better understand the algorithm performance (Line 164).

- Lines 169-203: it is a long paragraph. Perhaps it could be divided at lines 183 and 191

→ It is done in the revised manuscript (Line 186 & Line 193).

Lines 213-214: do the three percentages correspond respectively to percentiles 5, 50 and 95?
 → Yes, they are according to the reviewer. The English writing of the revised manuscript has been polished by a native speaker. Thus, each sentence is expected to have a clear meaning in the revised manuscript (Line 214 – Line 216).

- Line 217: comparison, not compassion

→ This typo mistake is corrected in the revised manuscript (Line 218).

- Line 221 (section 5.1). Is the NA radiometer a pirheliometer? Is it pointing at nadir direction continuously? Does it really measure radiance, or irradiance? I think it would useful to have some more details about the instrumentation used here.

 $\rightarrow$  The description is elaborated in the revised manuscript as below (Line 223 - 229).

The broad-band radiance and irradiance of the shortwave spectral range  $(0.3 - 2.8 \ \mu\text{m})$  observed using a narrow-angle radiometer (EKO Instruments Co., Ltd., Japan; FOV: 5°) and a pyranometer (Kipp and Zonen, Netherlands; FOV: 180°), respectively, at Chiba (35.62°N, 140.10°E) every 20 s from December 2015 to December 2016 are used to evaluate the cloud properties observed by the sky radiometer. The narrow-angle radiometer observes the downwelling irradiance signals as voltage in a narrow FOV. The instrument was calibrated by the manufacturer in the laboratory, and the observed signals are converted into radiance (unit: W/m2/sr) by using the company provided calibration constant value. Because the narrow-angle radiometer faces upward, thus obtained radiance is from the zenith.

**- Line 236: "highly qualitative"?**

→ "qualitative enough" is used instead of "highly qualitative" in the revised manuscript (Line
 240).

Section 5.2.: in order to better understand the improvement of the comparison respect to Kathri 2018, a short mention to the previous results using all the database would be useful
As suggested by the reviewer, a new paragraph is added in section 5.2 (Line 283 – Line 289).

Although the qualitive information reported by Khatri et al. (2018) and the comparisons in Figures 7 and 8 of this study are similar, there are differences in Figures 7 and 8 with the comparison plots shown in Khatri et al. (2018). The application of data screening criteria in this study generally screened out data with large differences between the sky radiometer and satellite sensors. These large differences in the previous comparison probably arose from the different FOVs of the satellite sensor and sky radiometer, while observing inhomogeneous clouds. Thus, the comparison results presented in this study by addressing the cloud inhomogeneity problem more logically should give more accurate and refined information than those presented in Khatri et al. (2018).

**Figures: - Line 270: Figures 7a and 7b**

 $\rightarrow$  The typo mistake is corrected in the revised manuscript (Line 272).

**- Line 276: Figures 8a and 8b**

 $\rightarrow$  The typo mistake is corrected in the revised manuscript (Line 277).

- Figure 4: I understand that figures a, b, c correspond to channels 870, 1020 and 1627. But I do not understand what are zenith and azimuth angles respectively. If both zenith and azimuth

results are the same for figures a and b, but are slightly different for figure c, please state that zenith and azimuth angles are represented with different colors.

The zenith and azimuth angles are exactly same for Figs. 4a, 4b, and 4c. Since the transmittance changes with the solar position, we choose solar zenith and azimuth angles of  $30^{\circ}$  and  $0^{\circ}$  as representative to investigate how the transmittances of 0.87, 1.02, and 1.627 µm change with the changes of COD and CER. Therefore, the solar zenith and azimuth angles are indicated in the caption.

If Fig. 4c is viewed very carefully, the contour lines denoted by different colors for 2 < COD< 5 and CER > 4 µm appear again for ~2

Figure 6: Scatterplot of broad-band radiances and irradiances observed with a narrow-angle radiometer and a wide-angle pyranometer at Chiba (35.62°N, 140.10°E) during January–March 2016. The solid line represents  $y = 2\pi x$ .

The solid line hardly suggests the overestimation from pyranometer as data points are scattered in both sides of a solid line. On the other hand, it indicates the asymmetric distribution of radiance.

**Technical corrections:**

**L28 "Foster" -> "Forster"**

 $\rightarrow$  It is corrected in the revised manuscript (Line 28).

**L59 "MCBride" -> "McBride"**

 $\rightarrow$  It is corrected in the revised manuscript (Line 56).

**L120 "(r)" rather than "(r2)"**

 $\rightarrow$  It is done in the revised manuscript (Line 115).

L163 "priory" -> "a priori"

 $\rightarrow$  It is corrected in the revised manuscript (Line 167).

L169 Fig. 3b, 3e, 3h, and 3k The style of right axis should be same as the other panels.

→ They are done in the revised manuscript (Page 15).

**L236 "qualitative" -> "quantitative" ?!**

 $\rightarrow$  It is qualitative (Line 240).

**L270 "Figures 6(a) and 6(b)" -> "Figures 7a and 7b"**

 $\rightarrow$  They are corrected in the revised manuscript (Line 272).

**Figure 7, Legends: Please omit the dual markers for each site.**

→ They are done in the revised manuscript (Page 18 & page 19).

**L276 "Figs. 7(a) and 7(b)" -> "Figs. 8a and 8b"**

 $\rightarrow$  They are corrected in the revised manuscript (Line 277).

| ĺ | 1      |                                               |       |                        |
|---|--------|-----------------------------------------------|-------|------------------------|
|   | 1
2 |                                               |       |                        |
|   | 2      |                                               |       |                        |
|   | 3      |                                               |       |                        |
|   | +
5 |                                               |       |                        |
|   | 5      |                                               |       |                        |
|   | 7      |                                               |       |                        |
|   | 8      |                                               |       |                        |
|   | 9      |                                               |       |                        |
|   | ,      |                                               |       | Formattad: Font: 24 nt |
|   | 10     | Changes made by authors to address the | $\gg$ | Formatted: Centered    |
|   |        |                                               |       |                        |
|   | 11     | reviewers' comments                           |       |                        |
|   | 12     |                                               |       |                        |
|   | 13     |                                               |       |                        |
|   | 14     |                                               |       |                        |
|   | 15     |                                               |       |                        |
|   | 16     |                                               |       |                        |
|   | 17     |                                               |       |                        |
|   | 18     |                                               |       |                        |
|   | 19     |                                               |       |                        |
|   | 20     |                                               |       |                        |
|   | 21     |                                               |       |                        |
|   | 22     |                                               |       |                        |
|   | 23     |                                               |       |                        |
|   | 24     |                                               |       |                        |
|   | 25     |                                               |       |                        |
|   | 26     |                                               |       |                        |
|   | 27     |                                               |       |                        |
|   | 28     |                                               |       |                        |
|   | 29     |                                               |       |                        |
|   | 30     |                                               |       |                        |
|   | 31     |                                               |       |                        |
|   | 32     |                                               |       |                        |
|   |        |                                               |       |                        |
|   |        | 1                                             |       |                        |

**Retrieval of cloud properties from sky radiometer observed spectral zenith radiances**

**35**

- 36 Pradeep Khatri1, Hironobu Iwabuchi1, Tadahiro Hayasaka1, Hitoshi Irie2, Tamio Takamura2,
- 37 Akihiro Yamazaki3, Alessandro Damiani2, Husi Letu4, Qin Kai5
- 38

[revised manuscript text omitted]

196
$$J = (\mathbf{x} - \mathbf{x}_a)^T \mathbf{S}_a^{-1} (\mathbf{x} - \mathbf{x}_a) + [\mathbf{y} - \mathbf{F}(\mathbf{x}, \mathbf{b})]^T \mathbf{S}_y^{-1} (\mathbf{y} - \mathbf{F}(\mathbf{x}, \mathbf{b})] , \qquad (2)$$

where x is a state vector,  $x_a$  is an a priori vector,  $S_a$  and  $S_y$  are error covariance matrices for the a priori and measurement, respectively, y is the measurement vector, F is the forward model, and b is the model parameter vector (ancillary data). The terms x, y, and b are defined as:

200
$$\mathbf{x} = \begin{pmatrix} ln\tau \\ lnr_e \end{pmatrix}, \ \mathbf{y} = \begin{pmatrix} lnT_{1.627} \\ lnT_{1.02} \\ lnT_{0.87} \end{pmatrix}, \text{ and } \ \mathbf{b} = \begin{pmatrix} W \\ A_{1.627} \\ A_{1.02} \\ A_{0.87} \end{pmatrix},$$

201 where  $\tau$  and  $r_e$  represent COD and CER, respectively; W and  $A_{\lambda}$  represent PWC and surface albedo at wavelength  $\lambda$ , 202 respectively. Both  $S_a$  and  $S_y$  are assumed to be diagonal matrices.  $x_a$  and the diagonal elements of  $S_a$  are determined 203 from one-year data of water cloud properties observed over Japanese SKYNET sites by Advanced Himawari Imager 204 (AHI) sensor onboard Himawari-8, a Japanese geostationary satellite. The diagonal terms for  $S_y$  are determined based 205 on simulation of perturbations in T(A) generated from 300 random gaussian noises of error sources discussed in section 206 4. The Santa Barbara DISORT Atmospheric Radiative Transfer (SBDART) model (Ricchiazzi et al., 1998) is used for 207 forward modelling, and the Levenberg-Marquardt method is used to minimize the cost-function. The total number of 208 iteration is set to 50. If the solution does not converge within 50 iterations, the analysis is discarded. As highlighted in 209 section 1 and shown in section 4, transmittance signals may not always be characterized by a unique COD and/or CER. 210 As a result, initial values of COD and CER used for iteration can keep a great importance while searching the plausible 211 set of COD and CER. To address this important issue, we first approximate the initial values of COD and CER to start 212 the iteration. It is done by searching a set of COD and CER through a comparison of observed  $T_{1.02}/T_{1.627}$  and  $T_{1.02}$  with 213 LUT of corresponding values modeled for COD ranging from 1 - 64 and CER ranging from  $2 - 32 \mu m$  at step of 1. 214  $T_{1.02}/T_{1.627}$  decreases with the increase of COD; whereas when COD increases,  $T_{1.02}$  increases first until reaching the 215 peak value, and then starts to decrease. Thus, simultaneous use of T102/T1627 and T102 can help one to figure out the 216 range of COD and CER, in which the true values are likely to fall. A set of COD and CER that can generate a least root 217 mean square difference between observed and modeled values is used as initial values in the iteration.

**219 4 Retrieval error**

218

In order to understand the performance of the proposed algorithm for different types of input data (transmittance\*/
 and ancillary data), retrieval errors are calculated by assuming errors on them. The retrieval errors are calculated for
 COD and CER in the intervals of 1 - 64 and 2 - 32 µm respectively, with steps of 1. The simulations are performed for

**Deleted:**

**Deleted:** calculated for COD of 1 - 64 and CER of 2 - 32um at an interval of 1 for each 226 
[revised manuscript text omitted]
 broad-band irradiance and broad-band radiance of spectral range mentioned above are observed by 807 instruments manufactured by Kipp and Zonen, Netherland and EKO Instruments Co., Ltd., Japan, respectively. The 308 latter can observe the downwelling irradiance signals in terms of voltage at a very narrow FOV. The instrument was 309 calibrated by a manufacturer in the laboratory, and by using such calibration constant value, it is possible to convert the 310 observed signals into radiance of unit W/m2/sr. As the sensor faces upward and the instrument's FOV is narrow, the 311 signal measured by this instrument can be regarded as the downwelling radiance of zenith direction. The cloud 312 properties from the sky radiometer are combined with MODIS observed surface albedo and radiosonde observed PWC 313 to calculate the counterparts of observation. A comparison is done for an average of 5 minutes observation of solar 314 radiation that centers the sky radiometer observation time. Figures 5(a) and 5(b) show the comparison for broad-band 315 radiance and irradiance, respectively. For reference, such comparison is done also for modeled values using cloud 316 properties of AHI instead of sky radiometer. They are shown in Figs. 5(c) and 5(d) for broad-band radiance and 317 irradiance, respectively.

318 Firstly, one can see a strong (weak) correlation between modeled and observed values for broad-band radiance 319 (irradiance) when cloud properties from sky radiometer are used. One the other hand, the correlation between modeled 320 and observed values for broad-band radiance (irradiance) are weak (strong) for AHI cloud properties. Rather than 321 pyranometer, the NA radiometer observed data can best describe the quality of sky radiometer cloud properties because 322 of narrow FOV. A very good agreement noted in Fig. 5(a) with correlation coefficient (r) as strong as 0.93 suggests that 323 sky radiometer based cloud properties (at least COD) are qualitative enough. As the contribution of COD is dominant 324 over CER in broad-band solar radiation (Khatri et al., 2018), Fig. 5(a) alone cannot justify the quality of retrieved CER. 325 At the same time, the relatively poor agreement for irradiance comparison shown in Fig. 5(b) can be described due to 326 significantly different FOV of sky radiometer and pyranometer. It is because the surface observed solar radiation can be 327 drastically different depending on the instrument FOV. As an example, Fig. 6 shows the scatter plot between 328 broad-band irradiance and radiance observed by pyranometer and NA radiometer, respectively at Chiba during January 329 - March, 2016. The correlation between them is very poor. One of the important factors that deteriorates the correlation 330 between them is the cloud horizontal inhomogeneity. This can plausibly explain the poor agreement in Fig. 5(b) despite 331 reasonably accurate retrieval from the sky radiometer as evidenced by Fig. 5(a). On the contrary, AHI cloud properties 332 are average (or representative) values of specific coverage, i.e., pixel (e.g., 1km x 1km). As a result, irradiances 333 modeled from AHI cloud properties become closer with observed irradiance than those modeled from sky radiometer 334 cloud properties. It is because the sky radiometer observed cloud can be just a small portion of a pixel containing 335 horizontally inhomogeneous clouds.

336

337 5.2 Satellite cloud products

(Formatted: Superscript

339 As part of validating water cloud products of MODIS and AHI using surface radiation data, Khatri et al. (2018) 340 compared water cloud properties retrieved from sky radiometer with those of MODIS and AHI observations for three 341 observation sites of SKYNET: Chiba, Hedo-misaki, and Fukue-jima for the period of October, 2016 to December, 2017. 342 They further used surface irradiance data, and found that the validation results using sky radiometer and surface 343 irradiance data are qualitatively same. A fairly good (poor) agreement was shown for COD (CER) between sky 344 radiometer and satellite products in Khatri et al. (2018), They compared sky radiometer results with results of 345 collocated satellite pixels by selecting samples with time difference less than 1.25 minutes, which is half of temporal 346 resolution of AHI observation over Japan, and the distance between the pixel center and the observation site less than 1 347 km, and further doing parallax correction for satellite products.

348 It is learnt from section 5.1 that the inhomogeneous clouds and/or broken clouds contained within the satellite 349 pixels are major obstacles in quality assessment of satellite products using sky radiometer results and vice versa. This 350 section attempts to improve our understating regarding the quality of sky radiometer products by using satellite products 351 effectively. For this purpose, we prepare samples for the comparison by addressing the cloud inhomogeneity related 352 problem in a logical way with available information at hand. If surface irradiance calculated from sky radiometer cloud 353 properties agree well with that observed at surface, the effective COD of actual inhomogeneous clouds may be 354 represented by a sky radiometer COD. Here, effective COD refers to COD of assumed plane-parallel homogenous 355 cloud layers which can produce irradiance equivalent to that produced by actual inhomogeneous clouds, i.e., measured 356 irradiance. Note that the satellite cloud properties retrieved from reflected signals assume clouds as plane-parallel 357 homogenous layers. The sky radiometer cloud properties that generate surface irradiances equivalent to observed values 358 by differing not more than 1% are selected to compare with satellite cloud properties. Figures 7(a) and 7(b) show the 359 comparison of sky radiometer CODs with MODIS and AHI values, respectively for sites and period same to Khatri et al. 360 (2018). The COD agreement is fairly good. The results are qualitatively same for both MODIS and AHI, by showing r361 values of ~0.6 and ~0.7 and RMSE values of ~13 and ~10 for MODIS and AHI, respectively. Despite several 362 differences between sky radiometer and satellite products from both observation and retrieval perspectives, a fairly 363 good agreement indicates that they can have similar response towards thin and thick clouds, though the COD value may 364 not be exactly same. Similarly, Figs. ¿(a) and ¿(b) show the comparison of sky radiometer CERs with MODIS and AHI 365 values, respectively. The water absorbing wavelengths corresponding to MODIS and AHI are 2.1 µm and 3.79 µm, 366 respectively. The CERs between sky radiometer and satellite sensors are poorly correlated. One can see r less than 0.12 367 and RMSE of  $\sim$ 7 µm for both satellite sensors. Such a poor correlation can be mainly due to the fact that satellite 368 sensors using reflected signals are highly sensitive towards cloud top layers (Platnick, 2000), whereas the sky 369 radiometer is sensitive to whole cloud layers.

Though qualitive information revealed from Khatri et al. (2018) and comparison results of Figures 7 and 8 of this
 study are in general the same, there are some differences in Figures 7 and 8 with respect to comparison plots shown in
 Khatri et al. (2018). Application of data screening criteria as mentioned above generally screened out those data that
 had considerably large difference between sky radiometer and satellite sensors. Such large difference in the previous

[revised manuscript text omitted]